# Modeling the Effects of Cypermethrin Toxicity on Ovalbumin-Induced Allergic Pneumonitis Rats: Macrophage Phenotype Differentiation and p38/STAT6 Signaling Are Candidate Targets of Pirfenidone Treatment

**DOI:** 10.3390/cells12070994

**Published:** 2023-03-24

**Authors:** Ahmed A. Morsi, Eman Mohamed Faruk, Mysara Mohamed Mogahed, Bodour Baioumy, Asmaa Y. A. Hussein, Rabab Shaban El-shafey, Ezat A. Mersal, Ahmed M. Abdelmoneim, Mohammed M. Alanazi, Amal Mahmoud ElSafy Elshazly

**Affiliations:** 1Department of Histology and Cell Biology, Faculty of Medicine, Fayoum University, Fayoum 63511, Egypt; 2Anatomy Department, College of Medicine, Umm Al-Qura University, Makkah 24382, Saudi Arabia; emkandel@uqu.edu.sa; 3Department of Histology and Cytology, Faculty of Medicine, Benha University, Benha 13511, Egypt; 4Department of Internal Medicine, Faculty of Medicine, Benha University, Benha 13511, Egypt; 5Department of Anatomy and Embryology, Faculty of Medicine, Benha University, Benha 13511, Egypt; 6Forensic Medicine and Clinical Toxicology Department, Faculty of Medicine, Benha University, Benha 13511, Egypt; 7Biochemistry Department, Faculty of Science, Assiut University, Assiut 71515, Egypt; 8Department of Basic Medical Sciences, Vision Colleges, Riyadh 11451, Saudi Arabia; 9Physiology Department, Faculty of Medicine, Fayoum University, Fayoum 63511, Egypt; 10Department of Pharmaceutical Chemistry, College of Pharmacy, King Saud University, P.O. Box 2457, Riyadh 11451, Saudi Arabia

**Keywords:** asthma exacerbation, CD86/CD206 immunohistochemistry, environmental triggers, p38/STAT6, Th2/Th1 inflammation

## Abstract

Although the classic form of asthma is characterized by chronic pneumonitis with eosinophil infiltration and steroid responsivity, asthma has multifactorial pathogenesis and various clinical phenotypes. Previous studies strongly suggested that chemical exposure could influence the severity and course of asthma and reduce its steroid responsiveness. Cypermethrin (CYP), a common pesticide used in agriculture, was investigated for the possible aggravation of the ovalbumin (OVA)-induced allergic pneumonitis and the possible induction of steroid resistance in rats. Additionally, it was investigated whether pirfenidone (PFD) could substitute dexamethasone, as an alternative treatment option, for the induced steroid resistance. Fifty-six male Wistar albino rats were randomly divided into seven groups: control, PFD alone, allergic pneumonitis, CYP alone, allergic pneumonitis/CYP-exposed, allergic pneumonitis/CYP/dexamethasone (Dex), and allergic pneumonitis/CYP/PFD-treated groups. Allergic pneumonitis was induced by three intraperitoneal OVA injections administered once a week, followed by an intranasal OVA instillation challenge. CYP (25 mg/kg/d), Dex (1 mg/kg/d), and PFD (100 mg/kg/d) were administered orally from day 15 to the end of the experiment. Bronchoalveolar lavage fluid (BALF) was analyzed for cytokine levels. Hematoxylin and eosin (H&E) and periodic acid Schiff (PAS)-stained lung sections were prepared. Immunohistochemical identification of p38 MAPK and lung macrophages was performed. The inflammatory/oxidative status of the lung and PCR-quantification of the STAT6, p38 MAPK, MUC5AC, and IL-13 genes were carried out. The allergic pneumonitis-only group showed eosinophil-mediated inflammation (*p* < 0.05). Further CYP exposure aggravated lung inflammation and showed steroid-resistant changes, p38 activation, neutrophil-mediated, M1 macrophage-related inflammation (*p* < 0.05). All changes were reversed (*p* < 0.05) by PFD, meanwhile not by dexamethasone treatment. Pirfenidone could replace dexamethasone treatment in the current rat model of CYP-induced severe steroid-resistant asthma via inhibiting the M1 macrophage differentiation through modulation of the STAT6/p38 MAPK pathway.

## 1. Introduction

Asthma is a chronic disabling respiratory disorder associated with a negative impact on the affected individuals’ socioeconomic status and quality of life. The global number of asthmatic individuals was reported to be increasing, reaching 300 million cases, including all age groups. The annual death rate due to asthma was reported to be about 250 thousand cases/year [1].

Microstructural and functional changes in the asthmatic patients’ lungs are cardinal signs attributed to T helper2 (Th2)-mediated inflammatory response due to Th1/Th2 imbalance [2]. Two clinical asthma variants (Th2 high and Th2 low) are known based on the clinical presentation and biochemical assays. Th2-high asthma is the classic allergic asthma manifested by high Th2-related cytokine (IL4, 5, and 13), eosinophils recruitment, goblet cell proliferation, and mucous hypersecretion. The Th2-low variant is usually a severe form, triggered by environmental pollutants, and manifested by high Th1/Th17-related cytokine levels, neutrophil recruitment, and steroid resistance [3].

The development of asthma is controlled by different regulators and transcription factors whose targeted therapy could be a viable option for treatment. The signal transducer and activator of transcription (STAT) family and mitogen-activated protein kinase (MAPK) are common molecular pathways in allergic asthma [4]. STAT6 activation mediates the classic Th2-high asthma response [5], whereas p38 MAPK activation is usually related to severe Th2-low asthma [6].

Exposure to environmental triggers, such as insecticides in agriculture, can initiate an asthmatic response in susceptible individuals, called occupational asthma [7]. Cypermethrin (CYP) is a highly active, commonly used synthetic pyrethroid pesticide. They are prevalent in agriculture, replacing prohibited insecticides such as organophosphate. Being a lipophilic compound, it could affect biological membranes with cardiovascular, nervous, immune, and genetic toxicity [8]. Exposure may be occupational, household, or accidental based on an adventure background. CYP was proven to have a toxic effect on aquatic species, mammals, and birds [9].

Pirfenidone (PFD) is a synthetic pyridine derivative approved in many countries as a recommended agent in treating idiopathic pulmonary fibrosis (IPF). Anti-inflammatory and anti-fibrotic actions were suggested mechanisms through inhibiting cytokine release, such as TNF-alpha and IL-6, suppression of inflammatory cell infiltration, and enhancement of inflammasomes [10]. The lack of literature studies emphasizing the curative leverage potential of PFD against environmental pollutants-induced severe asthma caught the authors’ attention for investigating pirfenidone versus dexamethasone.

Chronic asthma patients are susceptible individuals whose allergy could be worsened by insecticide exposure due to immunological shifts and alterations [11]. Therefore, the aim of the study was, first, to investigate the possible exacerbation of severe allergic lung inflammation and the possible induction of steroid resistance, due to oral CYP exposure, in concurrently existing asthma in albino rats. The current experiment is unique, investigating asthmatic immunoreactivity in a combined OVA-sensitization/CYP exposure model in rats. Second, the authors hypothesized that pirfenidone might be a proper alternative, versus dexamethasone, in treating the possible induced steroid resistance due to CYP exposure, using histological and biochemical approaches.

## 2. Materials and Methods

### 2.1. Chemicals and Drugs

Ovalbumin, OVA (≥97% purity powder, catalog #: S7951), and carboxymethyl cellulose (CMC 1%, catalog #: 419273) were purchased from Sigma-Aldrich, St. Louis, MO, USA. Aluminum hydroxide (Al (OH)_3_ was purchased from ThermoFisher Scientific (Waltham, MA, USA) (catalog #: AC219130250). Cypermethrin (20% emulsion concentration) was purchased from Kafr El Zayat Company for pesticides and chemicals, Gharbia, Egypt. Pirfenidone (Pirfenex 200 mg tab) was manufactured by Cipla LTD., Taza Block, Sikkim, India. Dexamethasone (Deltasone 0.75 mg tab) was manufactured by Nile Company (Cairo, Egypt) for pharmaceutical industries in Egypt.

### 2.2. Preparation and Dosage of Pirfenidone, Dexamethasone, Ovalbumin, and Cypermethrin

Both pirfenidone (PFD) and dexamethasone (Dex) tablets were ground and suspended in 1% carboxymethyl cellulose. One pirfenidone tablet was dissolved in 4 mL CMC to obtain a final concentration of 50 mg/mL and was given in a dose of 100 mg/kg/d oral gavage for two weeks [10]. Two dexamethasone tablets were dissolved in 3 mL CMC to obtain a final concertation of 0.5 mg/mL and were given in a dose of 1 mg/kg/day orally for two weeks [12]. Ovalbumin (OVA) solution was prepared (emulsified in aluminum hydroxide (adjuvant)) as previously described by Tang et al. [13] Both OVA (200 µg/mL) and aluminum hydroxide (40 mg/mL) were dissolved in normal saline, each per 1 mL saline; so, 1 mL solution contained 100 µg OVA and 20 mg Al(OH)_3._ Cypermethrin (CYP) was dissolved in corn oil and administered via oral gavage, 25 mg/kg/d (1/10 of LD50%), in a total volume of 5 mL/kg/d [14].

### 2.3. Induction of Allergic Asthma

For sensitization and challenge of rats, the method of Tang et al. [13] was followed with minor modifications. Each rat received 100 µL of the prepared OVA solution, intraperitoneal (i.p.), on days 0, 7, and 14. One week later, on the 21st day and to the 28th day, the rats were challenged daily by intranasal OVA instillation (Figure 1). For the challenge, the animals were lightly anesthetized and received OVA solution, 40 mg/mL; 20 µL/rat in nostrils to the nasal cavity.

### 2.4. Animals

Wistar male albino rats (*n* = 56, 6-week-old, 170–190 g bodyweight) were purchased from the breeding unit of the experimental animals’ farm, Helwan, Cairo, Egypt. For adaptation, the animals were fed for one week in the Anatomy Lab, Faculty of Medicine, Benha University, Egypt, before the experiment. The animals were housed in suitable living conditions (ambient temperature of 23 ± 2 °C, 55% humidity, and 12 h light/dark cycle) and food/water were allowed ad libitum. The study protocol was approved by the Research Ethics Committee at the Faculty of Medicine, Benha University, Egypt (approval #: RC11-10-2022).

### 2.5. Study Groups

The animals were randomly divided into seven groups, eight rats each: The control, Pirfenidone (PFD) alone, allergic pneumonitis-only, cypermethrin (CYP) alone, allergic/CYP, allergic/CYP/Dex, and allergic/CYP/PFD groups. In the control group, the rats received normal saline containing Al(OH)_3_ only by i.p. injection and were challenged by intranasal saline instillation, oral CMC 1%, and corn oil. The vehicles were given in the same volume and at the same time point, similar to the experimental groups. Dexamethasone was used herein as a reference drug for comparison with pirfenidone. CYP, PFD, and Dex were given orally and simultaneously from day 15 to day 28 (Figure 1). CYP was given at 9 am, PFD at 3 pm, and Dex at 9 pm to avoid possible drug–drug interaction.

### 2.6. Blood Sampling

Twenty-four hours after the last treatment, terminal blood samples were obtained from the heart of the anesthetized animals, centrifuged (176× *g* for 10 min), and supernatant sera were collected and stored at −20 °C for later analysis of serum IgE.

### 2.7. Bronchoalveolar Lavage Fluid (BALF) Preparation

After blood sampling, the anesthetized rats were sacrificed by cervical dislocation. Three animals per group were used for BALF collection. The trachea was cannulated after opening the chest and dissecting thoracic organs. Both lungs were washed with 5 mL of normal saline through the cannulated trachea with a light massage. Washing was repeated twice; then, the BALF was withdrawn (volume recovery rate more than 75%) and centrifuged (12× *g* for 10 min). For cell counting, the pellets were resuspended in 200 µL PBS and cells were manually counted by a hemocytometer under light microscopy after Giemsa staining. At least 200 cells were counted at high magnification (×400) [15]. Additionally, TNF-α, IL13, INF-γ, and IL-17 were measured in the BALF’s supernatants using rat-specific ELISA kits, following the manufacturer’s guidelines. The sensitivity of the assayed cytokines was 9.38 pg/mL for TNF-α and IL13, 7 pg/mL for INF-γ, and 3.9 pg/mL for IL17.

### 2.8. Lung Tissue Sampling

In each animal group, both lungs were resected, rinsed in cooled saline to remove blood clots, and separated into left and right lungs. The left lungs were stored at −20 °C for later molecular study and biochemical analysis of tissue homogenates. The right lungs were fixed in formal saline for histological and immunohistochemical study. Lung lavage was not performed in animals whose lung samples were used for histology and immunohistochemistry.

### 2.9. Tissue Homogenate Analysis of TNF-α, MDA, SOD

The collected lung samples were homogenized and centrifuged at 4 °C (704× *g*, for 20 min). The obtained supernatants were frozen at −20 °C for later measurement of tumor necrosis factor-α (TNF-α), malondialdehyde (MDA), and superoxide dismutase (SOD) using ELISA assay. Rat-specific kits for MDA and SOD were purchased from MyBioSource, San Diego, CA, USA, catalog #: MBS738685 and MBS036924, respectively. For TNF- α, the kits were purchased from Elabscience, Houston, TX, USA, catalog #: E-EL-R2856.

### 2.10. PCR Assay for Quantification of STAT6, p38MAPK, MUC5AC, and IL-13 Gene Expressions

Initial homogenization of the collected lung samples was carried out. The total RNA was isolated using RNeasy Mini Kit (Cat No./ID: 74104, Qiagen, Germantown, MD, USA) and spectrophotometric (JENWAY, Long Branch, NJ, USA) determination of the RNA concentration was carried out at 260 nm. The mRNA was converted into cDNA using a PCR Master Mix kit (catalog #: 4440040, Applied Biosystems, Waltham, MA, USA). The PCR amplification/analysis was evaluated by StepOneSystem (RQ Manager 1.2, software v 2.1, Applied Biosystem, USA). The result of the targeted genes (*STAT6*, *p38*, *MUC5AC*, and *IL-13*) was normalized to an endogenous housekeeping gene GAPDH. The comparative 2^−∆∆Ct^ method was used for the calculation of the relative fold gene expression, as demonstrated by Livak & Schmittgen [16]. For polymerase activation/reverse transcriptase inactivation, the thermocycler was set at 45 °C, followed by 95 °C (for 15 and 5 min, respectively) [17]. For detection of the rat *STAT6*, *p38*, *MUC5AC*, and *IL-13* transcripts, the primer sequences (Table 1) were taken from NCBI GenBank at the following website: http://www.ncbi.nlm.nih.gov/tools/primer-blast (accessed on 1 January 2023). A melting temperature of 60–65 °C and a length of approximately 90–200 bp were considered when selecting the ideal primer pair.

### 2.11. Histological Procedure

The right lung of each rat per group was processed for paraffin section preparation after 48- hour-fixation in 10% formol saline at room temperature. The samples were dehydrated by passing through ascending grades of ethanol (70%, 90%, and 100%), cleared in xylene, infiltrated, and embedded in paraffin. For each paraffin block, five-micrometer cut sections were prepared for histological stains. Sections were deparaffinized, rehydrated in descending grades of ethanol, and manually stained with hematoxylin and eosin (H&E) and periodic acid Schiff (PAS) [18].

### 2.12. Immunohistochemical Identification of CD86, CD206, and p38 MAPK

The formalin-fixed, paraffin-processed cut sections of the lungs were mounted on positively charged glass slides. They were subjected to the sequential steps of the immunohistochemical assay [19] to identify p38 MAPK and CD markers of macrophage phenotypes (CD86, CD206). The primary antibodies (CD86, p38, and CD206) were all rabbit polyclonal antibodies purchased from ABclonal Inc., Woburn, MA, USA (catalog #: A2353, A14401, respectively). CD206 was purchased from ThermoFisher Scientific (catalog #: PA5-101657). After deparaffinization and rehydration, the sections were boiled in citrate buffer (Thermo Fisher Scientific, Epredia, catalog #: AP9003-125) for 10 min at pH 6 to retrieve the antigen, and were then cooled naturally. The slides were incubated with an H_2_O_2_ solution (15 min) to block endogenous peroxidase activity. Block the sections with the corresponding protein. Then, the lung sections were incubated with the primary antibodies overnight at 4 °C (at dilutions 1:200 for p38 and CD86, 1:100 for CD206). Each step mentioned was followed by PBS washing. For completing the immunoreaction, a rabbit-specific anti-polyvalent HRP/DAB ultravision detection system was used (TP-015-HD, Lab Vision™, Thermo Fisher Scientific). The 3,3-diaminobenzidine (DAB) chromogen and hematoxylin counterstaining were used.

The CD86 and CD206 immunopositivity was marked by brownish cytoplasmic coloration. For p38, positive nuclear and cytoplasmic reaction was identified. The positive controls were mouse spleen, mouse lung, and human heart for CD86, CD206, and p38, respectively. The negative control slides were arranged by skipping the step of the primary antibody. The slides were examined and photographed by a Nikon Eclipse 80i microscope (Nikon Corporation, Tokyo, Japan) provided with ToupCam^TM^ Xcam full HD camera (ToupTek Europe, Ultramacro Ltd., Suffolk, UK). Slide examination and photography were done in the Department of Anatomy and Embryology, Faculty of Medicine, Benha University, Egypt.

### 2.13. Morphometric Study

Blind image analysis was conducted by an expert in the field from whom the study groups’ information was withheld. This was performed using ImageJ software (Java; NIH, Bethesda, MD, USA), which was calibrated to analyze the captured photomicrographs [20]. The PAS-stained slides were used for calculating the goblet cell number (GCN); meanwhile, the H&E-staining was used for measurement of the eosinophil number (EN), neutrophil number (NN), perivascular inflammation, peribronchiolar inflammation, and alveolar wall thickness. For GCN measurement, the goblet cells were counted per five round cross-cut bronchioles in each section per group at ×200 magnification. The average goblet cells/bronchiole was obtained by dividing the total number by five, according to Tang et al. [21] with minor modifications. For eosinophil and neutrophil numbers, the peribronchial cells were counted in five bronchioles in each section, and the parenchymal cells were counted in five fields at ×200. The average EN or NN/field was calculated by dividing the obtained number by five [22]. The perivascular and peribronchiolar inflammation was assessed by counting the number of complete layers of mononuclear cells (MNCs) around blood vessels and bronchioles, respectively, as follows: 0 (absence or presence of few MNCs), 0.1–0.9 (incomplete layer), 1 (one complete layer), 2 (two complete layers), 3 (three complete layers), 4 (more than three complete layers) according to Wu et al. [23]. The immunoassay-based slides were used for measurement of the area percentage of p38 MAPK immunopositivity and counting the number of the CD86- and CD206-positive macrophages. The macrophage phenotype was counted for those located peribronchial and interstitially in 5 bronchioles/or fields.

### 2.14. Statistical Analysis

The normality check and even distribution of variables were initially carried out by applying Shapiro’s test. One-way analysis of variance (ANOVA) was used to detect the statistical significance, accompanied by post hoc Tukey’s test for multiple groups comparison. The measurements of CD86 and CD206 positive macrophages were carried out by two-way ANOVA followed by Tukey’s test for comparison between the groups and the two variables within each group. All data were expressed as mean ± SD. The data analyses were conducted using GraphPad Prism software, v8.0, for windows (GraphPad Software Inc., San Diego, CA, USA). Statistical significance was considered at *p* < 0.05. G *power software version 3.0.10 was used to determine the total sample size and the number of rats per group [24]. The initial setting of the software was carried out, and an a priori type of power analysis was selected, power 80, alpha level 5%, and a desired effect size 0.55, predetermined based on the parallel finding of similar studies and experience in the field area. Thus, eight rats per group were statistically adequate.

## 3. Results

### 3.1. Pirfenidone Reversed the Serum Level of IgE

The serum immunoglobulin E (IgE) level showed a remarkable elevation (*p* ˂ 0.05) in the allergic pneumonitis group, compared to the control IgE levels. CYP exposure of the allergic pneumonitis (allergic pneumonitis/CYP) group caused a significant decrease in the IgE level (*p* ˂ 0.05), compared to the allergic pneumonitis-only group. Pirfenidone treatment (allergic/CYP/PFD group) versus dexamethasone (allergic/CYP/Dex group) showed a marked decrease (*p* ˂ 0.05) in the IgE level when compared to the allergic/CYP group. No significant difference was noticed between the control and PFD-alone treatment (Table 2).

### 3.2. Pirfenidone Suppressed Inflammatory Cytokines, Eosinophilic and Neutrophilic Count in BALF

The inflammatory status was assessed by measuring TNF-α, INF-γ, IL13, IL17, and cell counts in bronchoalveolar lavage fluid (BALF) (Table 2). The four measured cytokines and eosinophil count showed a remarkable elevation (*p* ˂ 0.05) in the allergic pneumonitis group when compared to the corresponding control levels. CYP-alone exposure led to a significant elevation (*p* ˂ 0.05) in TNF-α and INF-γ levels compared to the control group. CYP exposure of the allergic pneumonitis (allergic/CYP) group shifted the cell count toward neutrophil (*p* ˂ 0.05) and caused a further significant increase (*p* ˂ 0.05) in the BALF levels of INF- γ and IL17 only when compared to the allergic pneumonitis-only group. Pirfenidone (allergic/CYP/PFD group) versus dexamethasone treatment (allergic/CYP/Dex group) showed a marked decrease (*p* ˂ 0.05) in all measurements when compared to the allergic/CYP group. No significant difference was noticed between the control and PFD-alone treatment.

### 3.3. Pirfenidone Alleviated Inflammation and Oxidative Stress in the Lung Tissues

MDA, SOD, and TNF-α were measured in the lung tissue homogenates to evaluate inflammation and the oxidant/antioxidant condition (Table 3). Both MDA and TNF-α levels showed a remarkable elevation (*p* ˂ 0.05) in the allergic pneumonitis and CYP alone groups when compared to the control group. CYP exposure of the allergic pneumonitis (allergic/CYP) group caused a further significant increase (*p* ˂ 0.05) in the MDA and TNF-α measurements when compared to the allergic pneumonitis-only group. Inversely, the SOD level was markedly decreased (*p* ˂ 0.05) in the allergic pneumonitis group, and a lower level (*p* ˂ 0.05) was observed in the allergic/CYP group. Pirfenidone treatment (allergic/CYP/PFD) group versus dexamethasone significantly ameliorated (*p* ˂ 0.05) the oxidative changes. No significant difference was noticed between the control and PFD-alone treatment.

### 3.4. Pirfenidone Modulated STAT6, p38MAPK, MUC5AC, and IL-13 Gene Expression

The PCR assay of lung samples was displayed in Figure 2. The *IL-13*, *MUC5AC*, *STAT6*, and *p38* mRNA expression showed a significant rise (*p* < 0.001) in the allergic pneumonitis group when compared to the control gene values. Furthermore, CYP exposure in the allergic pneumonitis rats resulted in more increase (*p* < 0.003) in *MUC5AC* and *p38* gene expression (Figure 2b,c, respectively); meanwhile, the *IL-13* and *STAT6* (Figure 2a,d, respectively) showed a further decrease (*p* < 0.05), in comparison to the allergic pneumonitis-only group. Pirfenidone treatment (allergic/CYP/PFD group) versus dexamethasone (allergic/CYP/Dex) group showed a significant reversal (*p* < 0.05) of the altered gene expression profiles.

### 3.5. Pirfenidone Improved the Histological Structure of the Lungs

Microscopic examination of the H&E-stained lung sections of both control (Figure 3a) and pirfenidone alone-treated (Figure 3b) groups revealed healthy histological structure of the rats’ lungs. Healthy alveoli were lined by numerous thin squamous cells, type I pneumocytes, and a few thick cubical cells, type II pneumocytes. The alveoli were separated by thin interalveolar septa containing blood capillaries. Bronchioles were lined by columnar cells with thin underlying smooth muscle layers.

The allergic pneumonitis group (Figure 3c,d) displayed thick interalveolar septa due to massive cellular infiltrates mainly eosinophils, interalveolar edema, and dilated congested blood capillaries (Figure 3c). Bronchioles showed swollen walls due to thickened smooth muscle layer, eosinophil infiltration, and hypertrophied-folded epithelium intervened by goblet cells (Figure 3d). The CYP-exposed group (Figure 3e) showed distorted alveoli with disorganized epithelial lining. Dilated blood vessels and diffuse vascular congestion were also observed, with interalveolar and intra-alveolar extravasation of red blood cells.

However, an aggravated effect was noted in the allergic/CYP-exposed group (Figure 3f,g). The lung sections showed more bronchial affection with narrowing and partial obliteration of the alveoli. The bronchiolar walls were thicker than the allergic-only group due to hypertrophy of the bronchiolar epithelium with numerous intervening goblet cells, thickened smooth muscle layer, and massive peribronchiolar mononuclear cellular infiltrations (Figure 3f). The lung alveoli showed more thickening of the interalveolar septa with neutrophil infiltration leading to obliterated lumina (Figure 3g). The allergic/CYP/Dex-treated group (Figure 3h) showed persistent histopathological changes with resistance to dexamethasone treatment. The allergic/CYP/PFD-treated group (Figure 3i) showed remarkable improvement in the lung fields with the resolution of the edema and inflammation.

The PAS reaction showed negative PAS staining in the control and PFD-alone groups (Figure 4a,b, respectively). Meanwhile, numerous PAS-positive goblet cells were scattered among the airway epithelium of the allergic pneumonitis-only group (Figure 4c). The CYP alone-exposed group showed scarce PAS-reacted goblet cells (Figure 4d). However, extensively distributed goblet cells were noted in the combined model (allergic/CYP-exposed rats) (Figure 4e), whereas their number dramatically decreased during PFD treatment (Figure 4g) versus dexamethasone (Figure 4f).

### 3.6. Pirfenidone Inhibited the Lung Immunoexpression of p38 MAPK

The alveolar epithelia and interstitial macrophages showed p38 immunoreactivity in the pulmonary tissues of the allergic (Figure 5c) and the CYP-exposed (Figure 5d) animal groups. The allergic/CYP-exposed group (Figure 5e,f) showed a remarkable p38 immunopositivity with a substantial expression in the alveolar and bronchial lining cells and interstitial macrophages. In the combined model, dexamethasone treatment (allergic/CYP/Dex group (Figure 5g) showed steroid resistance denoted by the persistence of the p38 immunopositivity, meanwhile PFD treatment (allergic/CYP/PFD group (Figure 5h) significantly inhibited the p38 immunoreaction.

### 3.7. Pirfenidone Modulated the Immunoexpression of Macrophage Markers, CD86 and CD206

Comparing the immunoexpression of the two macrophage markers in each group revealed substantial expression of the M1 macrophage marker (CD86) in the CYP alone-exposed animals (Figure 6d) and predominant CD206 expression in the allergic pneumonitis group (Figure 6j). Furthermore, CYP exposure in the allergic group (allergic/CYP group) shifted the macrophage polarization toward the M1 phenotype (Figure 6e). PFD treatment (Figure 6g,n) significantly inhibited both macrophage phenotypes, while not in dexamethasone treatment (Figure 6f,m), indicating steroid refractoriness. Taken together, the results indicated that pirfenidone is efficient in reducing both macrophage phenotypes (M1 and M2) in allergic/CYP-exposed lung tissues.

### 3.8. Pirfenidone Reversed the Histomorphometric Measurements in the Lung Tissues

The mean eosinophil number was significantly greater (*p* < 0.001) in the allergic group when compared to the control and CYP alone group and significantly decreased (*p* < 0.001) in allergic/CYP-exposed groups. Dexamethasone had no further effect in comparison to allergic/CYP-exposed groups; however, PFD treatment had a significant (*p* < 0.001) effect (Figure 3j). In contrast, the mean neutrophil count was predominant (*p* < 0.001) after CYP exposure in the allergic rats (allergic/CYP-exposed), in comparison to the allergic pneumonitis-only group (Figure 3k), and their number was reversed (*p* < 0.05) by PFD but not dexamethasone treatment. Moreover, the allergic/CYP-exposed group showed a marked increase (*p* < 0.05) in the mean alveolar wall thickness (Figure 3l), mean goblet cell number (Figure 4h), and inflammation (Table 4) when compared to the allergic pneumonitis group. The mean area percentage of p38 MAPK immunoreactivity (Figure 5i) showed a marked (*p* < 0.001) p38 expression in the CYP-exposed allergic rats when compared to the allergic pneumonitis-only group. The p38 immunoexpression was reversed (*p* < 0.001) by PFD versus Dex treatment. Regarding the dominant macrophage phenotype (Figure 6o,p), the lung CD206 immunoexpression (an indicator of M2 phenotype) was significantly predominant (*p* < 0.001) in the allergic pneumonitis-only group. CYP exposure in allergically sensitized rats facilitated the macrophage shift (*p* < 0.001) toward the M1 phenotype, as indicated by the lung CD86 dominant expression, which was not revered by dexamethasone. PFD-versus-dexamethasone treatment improved (*p* < 0.05) all the lung parameters and reduced (*p* < 0.001) both M1 and M2 macrophage phenotypes.

## 4. Discussion

Utmost patients with classic asthma show a better response to conventional steroid therapy. Meanwhile, other patient classes exhibit steroid resistance and persistence of asthma, which entails a high dose and long-term glucocorticoid treatment [25]. The current findings showed that cypermethrin, a widely used pesticide with common indoor and outdoor exposures [26], aggravated lung inflammation and was associated with dexamethasone irresponsiveness in the OVA-immunized rats. Notably, PFD treatment improved the histobiochemical pulmonary alterations and could replace dexamethasone treatment, bypassing steroid resistance. In accordance, Pelaia et al. [27] demonstrated that environmental triggers such as cigarette smoke, air pollutants, and lung infections could initiate severe Th2 low steroid-resistant asthma in susceptible individuals.

In the OVA-sensitized group, the histological and biochemical findings indicated classic eosinophilic asthma mediated by Th2 cell-dependent airway response (Th2-high inflammation). The results revealed upregulated IL13 gene/protein expression, eosinophilic pneumonitis, and upregulated *MUC5AC* gene expression coupled with goblet cell hyperplasia. Furthermore, *STAT6* gene expression was markedly upregulated, which was consistent with the findings mentioned earlier. In agreement, Chu et al. [28] mentioned a critical role for STAT6-associated pathways in the Th2 airway immune response and IL4/IL13 signaling. In the CYP-alone exposed group, the results displayed histological and biochemical features of a chemical-induced pneumonitis consistent with Pelaia et al. [27], who showed parallel pulmonary alterations.

Neutrophilic pneumonitis is a prominent feature of Th2 low-type mediated asthma and is commonly associated with severe asthma and steroid resistance [3]. In the current study, CYP-exposure of the allergic rats (allergic/CYP-exposed group) resulted in an aggravation of the allergic airway inflammation induced by OVA. Neutrophil recruitment, mononuclear cell infiltrates, and extensive goblet cells were prominent in the histological stains. In accordance, the BALF levels of Th1-related cytokine (INF-γ) and Th17-associated cytokine (IL17) were predominant. Marked lung oxidant/antioxidant imbalance was evident. Additionally, CYP further exaggerated *MUC5AC* gene upregulation, synergized *p38 MAPK* gene/protein expression, inhibited *STAT6* gene expression exerting STAT6-to-MAPK signaling shift and M2-to-M1 macrophage phenotype transition as demonstrated by the PCR findings and the immunohistochemical (IHC) observations. We speculated p38 MAPK/STAT6 inverse relationship, where the activated p38 MAPK interacted with STAT6 with inhibition of the latter. This assumption agreed with Athari [4], who declared STAT6 activation blockage by interfering with the STAT6-MAPK interaction with p38.

Lung macrophages have intrinsic anti-inflammatory activity maintaining lung homeostasis via a balanced M1/M2 response. On lung injury, the resident macrophages can skew toward a proinflammatory phenotype based on their controlled switching ability between functional phenotypes [29]. In this study, the macrophage CD markers revealed the predominance of CD206 immunopositive M2 phenotype in the allergic pneumonitis-only group and switched to CD86 positive M1 phenotype after CYP exposure. Macrophage transformation is controlled by different transcription factors; the most important are STATs family. While STAT6 directs M2 polarization, STAT1 regulates M1 switch [5]. This notion emphasized our results of the STAT6-associated M2 polarization in the allergic pneumonitis-only group and the possible p38-related M1 polarization in the combined model group. The p38-induced STAT6 inhibition might inhibit M2 phenotype during CYP exposure leading to M1 polarization shift and subsequent neutrophilic inflammation as suggested by Draijer et al [30].

Unlike the allergic pneumonitis-only group, which is classically influenced by the Th2 response, the airway neutrophilic inflammation observed in the allergic/CYP-exposed group might be attributed to further oxidative stress induced by CYP exposure that shifts the immune reactivity to INF-γ/Th-17 axis and p38 MAPK activation. Our suggestion was corroborated by Bao et al. [6], who reported a combined role of p38 MAPK and oxidative stress pathways in ozone-induced augmentation of lung inflammation in allergic mouse model. Therefore, we hypothesized an instantaneous inhibition of both oxidative stress and p38 MAPK pathways as a postulated mechanism of action for the pirfenidone ameliorative effects in the current combined model of CYP-exposure/OVA-sensitization. This theory was supported by Li et al. [31] and Fois et al. [32], who reported anti-oxidative and p38 inhibitory activities for PFD as a basis for its antifibrotic effects.

Compared with the combined model group, p38 gene/protein expression displayed a significant decrease with PFD treatment, meanwhile not with dexamethasone treatment. Therefore, it was hypothesized that p38 signaling cascade might be involved in CYP-induced steroid-resistant asthma. The results revealed that CYP exposure increased p38 MAPK gene/protein expression in the structural cells of the lung, which was successfully suppressed by PFD. Despite the lack of PFD literary data on experimental models of allergic asthma, two previous studies [33,34] suggested a curative potential for PFD in the classic asthma phenotype by suppressing the BALF’s inflammatory cytokine and platelet-derived growth factor, inhibiting TGF-β1 expression, and reducing the goblet cell number & subepithelial fibrosis.

p38 upregulation, in the current study, was suggested as a reason for dexamethasone resistance. In agreement, Zeyen et al. [35] reported crosstalk between glucocorticoid receptors (GRs) and p38 MAPK, and the activation of the latter could impede GRs function and induce steroid resistance. Thus, the data propose PFD as a valid alternative to dexamethasone in steroid-resistant cases by modulating M1 macrophage polarization-associated p38 MAPK signaling pathway. Further studies are warranted to explore whether PFD might increase the pulmonary sensitivity to steroid treatment in steroid-resistant asthma.

The marked INF-γ upregulation in the combined model group might be an explanation for redirecting the inflammatory pathway, STAT6 downregulation, p38 MPAK activation, and hence, M2-to-M1 polarization shift. In their latest review, Abdelaziz et al. [36] supported our assumption. They reported that INF-γ (Th1-associated cytokine) could repress the Th2 cell lineage development, negatively affect STAT6 signaling, and activate M1 phenotype macrophage switch.

The current study was the first work to introduce the combined rat model of OVA-sensitization/CYP exposure. In addition, the study presented PFD as a potent alternative for dexamethasone in alleviating the augmented pulmonary immune response and the resistance to steroid treatment induced by CYP in allergic pneumonitis rats. However, the study had a few limitations. First, it did not introduce both PFD and Dex combined in one group to demonstrate whether PFD could fend off steroid resistance, and so, further information would be provided by future studies. Second, the quantification and characterization of the BALF’s macrophages were limited in the current study. Additionally, the M2-to-M1 phenotype change needs further identification in association with studied markers. Third, the STAT6-to-p38 MAPK shift and signaling pathway need further investigations via studying protein phosphorylation using Western blot; otherwise, an in vitro experiment could be designed for investigating the PFD effect on STAT6/p38 MAPK pathway and its role in M1 macrophage differentiation.

In summary, CYP cannot initiate allergic asthma alone; it may provoke asthma as an environmental trigger with allergens through a mixed Th1/Th2-mediated airway inflammation. The findings showed that CYP exposure exaggerated the airway inflammation induced by ovalbumin sensitization and produced a full-blown picture of Th2 low severe asthma featured by steroid resistance, high INF-γ/IL17, neutrophilic pneumonitis, M1 phenotype predominance, and abundance of the epithelial goblet cells with mucous hypersecretion. The interaction between cytokines (IL13, INF-γ) and transcription factors (STAT6, p38 MAPK) played a critical role in this immune response. The present study presented CYP as an environmental trigger that could initiate severe dexamethasone-resistant asthma in sensitized rats. Furthermore, pirfenidone was suggested as a viable treatment option that protected against CYP-induced steroid-resistant allergic pneumonitis via M1 macrophage polarization suppression by modulating IL-13/STAT6 and INF-γ/p38 pathways.

## Figures and Tables

**Figure 1 cells-12-00994-f001:**
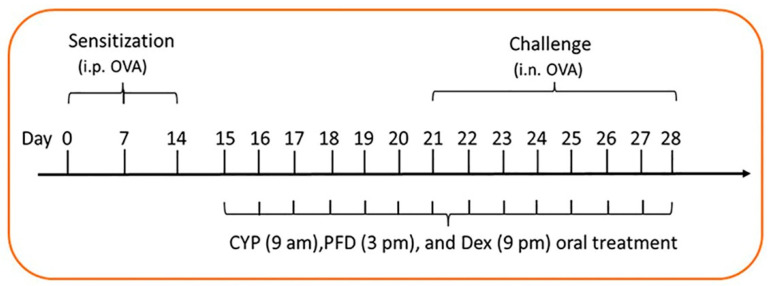
Diagrammatic representation of the different timepoints of the current experiment. i.p.: intraperitoneal, i.n.: intranasal, OVA: ovalbumin, CYP: cypermethrin, PFD: pirfenidone, Dex: dexamethasone.

**Figure 2 cells-12-00994-f002:**
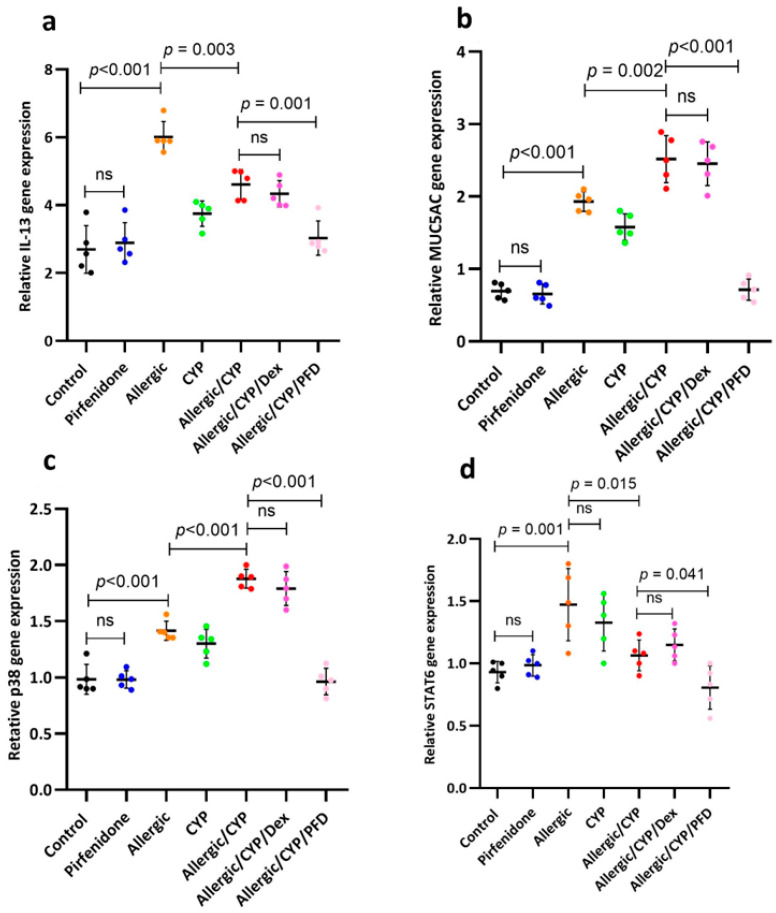
The effect of pirfenidone on the gene expression profile of IL-13 (**a**), MUC5AC (**b**), p38 (**c**), and STAT6 (**d**) genes determined by PCR in the CYP-exposed allergic rats at the end of the experiment. The data are shown as mean ± SD, *n* = 5 lung tissue samples. Horizontal lines indicate the statistical significance and the *p*-value between groups using one-way ANOVA (Tukey’s post hoc test). ns: nonsignificant.

**Figure 3 cells-12-00994-f003:**
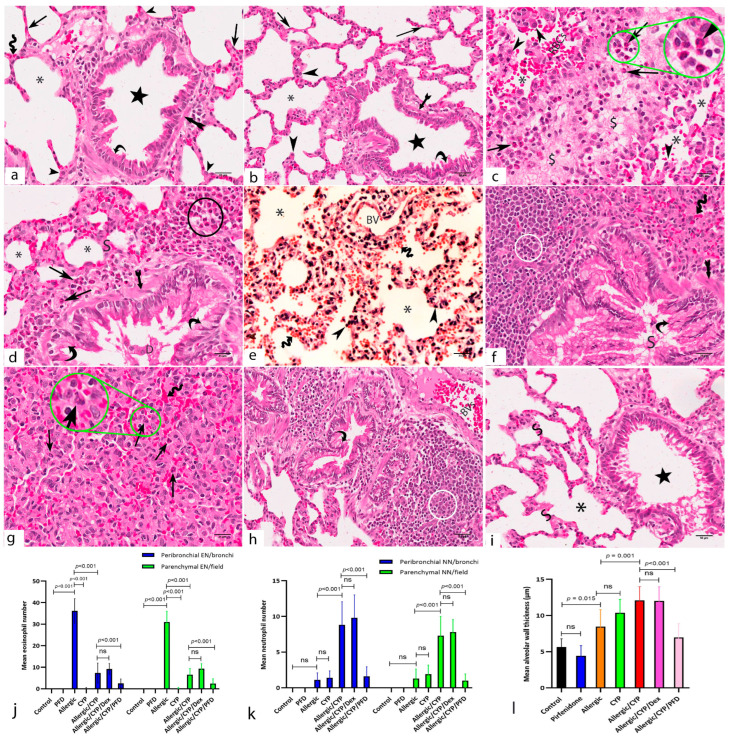
Photomicrographs of H&E-stained lung sections of the study groups. (**a**,**b**) control and pirfenidone alone-treated groups, respectively, show normal histological structure of the lungs. Alveoli (asterisks) are lined by thin type I (arrows) and thick type II (arrowheads) pneumocytes. Thin interalveolar septa are seen containing blood capillaries (zigzag arrows). Bronchioles (star) are lined by columnar cells (curved arrow) with thin underlying smooth muscle layers (arrow with tails). (**c**,**d**) Allergic pneumonitis group shows narrowed alveoli (asterisks) and thick interalveolar septa (S) infiltered by mononuclear inflammatory cells (circle), mainly eosinophils (arrows). Interalveolar edema (dollar sign) and massive interalveolar eosinophil infiltration (arrows) are prominent. Many alveoli are distorted with hypertrophic cell lining (arrowheads). Intra-alveolar red blood cells (RBCs) and vascular congestion are evident. The bronchioles show a thickened smooth muscle layer (arrows with tail), eosinophil infiltration (arrow), and hypertrophied folded epithelium with goblet cells (curved arrows). Detached cells and luminal mucous accumulation (D) are seen. (**e**) CYP-exposed group shows distorted alveoli (asterisks) with disorganized epithelial lining (arrowheads). Dilated blood vessels (BV) and diffuse vascular congestion (zigzag arrows) with extravasated blood cells are also seen. (**f**,**g**) Allergic/CYP-exposed group shows hypertrophic, highly folded bronchiolar epithelium with numerous goblet cells (curved arrows). Mucous (S) is accumulated intra-bronchiolar. The bronchiolar smooth muscle layer is thick (arrow with tail). Extensive mononuclear cellular accumulation is seen peribronchiolar (circle). The alveoli appear with obliterated lumina and show neutrophil infiltration (arrows) and congested capillaries (zigzag arrows). (**h**) Allergic/CYP/Dex-treated group shows the persistence of the cellular infiltration (circle) and goblet cell hyperplasia (curved arrows). (**i**) Allergic/CYP/PFD-treated group shows remarkable improvement of alveoli (asterisk), interalveolar septa (S), and bronchiolar (star) structures with the resolve of the edema and inflammation. (**j**–**l**) representative figures of the mean eosinophil number (EN), neutrophil number (NN), and alveolar wall thickness, respectively, *n* = 5. The significant difference and the *p*-value are shown between the group using post hoc Tukey’s test, one-way ANOVA. ns: nonsignificant. Scale bar: 50 µm (**a**,**i**); 25 µm (**b**–**h**).

**Figure 4 cells-12-00994-f004:**
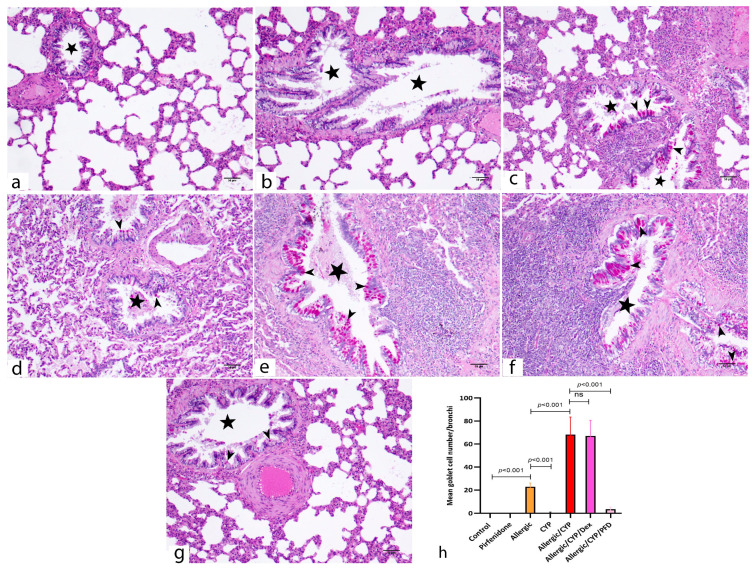
Photomicrographs of PAS-stained lung sections of the study groups. The PAS-positive goblets cells (arrowheads) appear as an intense magenta-red reaction among the bronchiolar (stars) epithelium. (**a**,**b**) control and PFD alone-treated groups, respectively, show negative PAS reactions. (**c**) The allergic pneumonitis group shows numerous PAS-positive goblet cells scattered among the bronchiolar epithelium. (**d**) CYP-exposed group shows scarce PAS-positive goblet cells. (**e**) The allergic/CYP-exposed group shows extensively distributed PAS-reacted goblet cells intervening the epithelial lining. (**f**) The allergic/CYP/Dex-treated group shows persistent numerous PAS-stained goblet cells. (**g**) The allergic/CYP/PFD-treated group shows very few PAS-positive goblet cells. (**h**) Representative figure of the mean goblet cell number per bronchiole in the studied groups at the end of the experiment, *n* = 5. The significant difference and the *p*-value are shown between the group using post hoc Tukey’s test, one-way ANOVA. ns: nonsignificant. Scale bar: 50 µm (**a**–**g**).

**Figure 5 cells-12-00994-f005:**
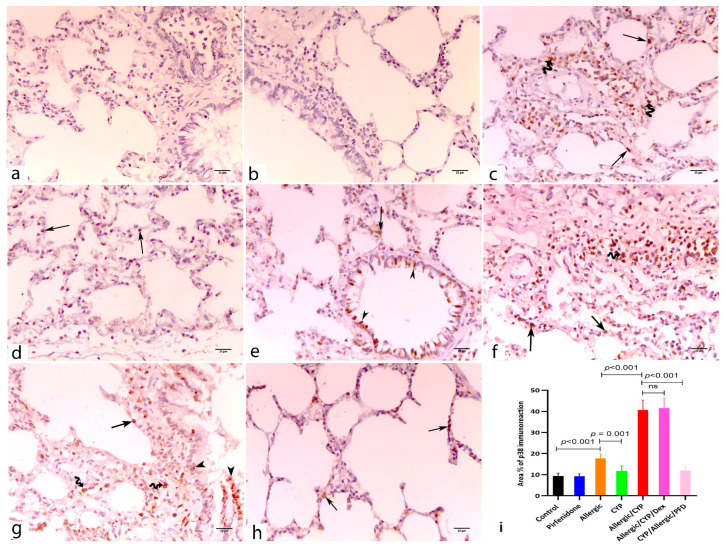
Anti-p38MAPK immunoassay lung sections of the study groups. (**a**,**b**) control and PFD alone-treated groups, respectively, show negative immunoreaction. (**c**) Allergic pneumonitis group shows p38 positive alveolar lining cells (arrow) and interstitial macrophages (zigzag arrows). (**d**) CYP-exposed group shows few p38 positive alveolar cells (arrows). (**e**,**f**) Allergic/CYP-exposed group shows extensive distribution of p38 immunopositivity staining the alveolar (arrows), bronchial lining cells (arrowheads), and interstitial macrophages. (**g**) Allergic/CYP/Dex-treated group shows the persistence of the p38 +ve alveolar (arrows) and bronchial cells (arrowheads), in addition to macrophages (zigzag arrow). (**h**) Allergic/CYP/PFD-treated group shows very few positive alveolar cells (arrows). (**i**) Representative figure of the mean area percentage of p38 immunopositivity among the study groups at the end of the experiment, *n* = 5. The significant difference and the *p*-value are shown using post hoc Tukey’s test, one-way ANOVA. ns: nonsignificant. Scale bar: 25 µm (**a**–**h**).

**Figure 6 cells-12-00994-f006:**
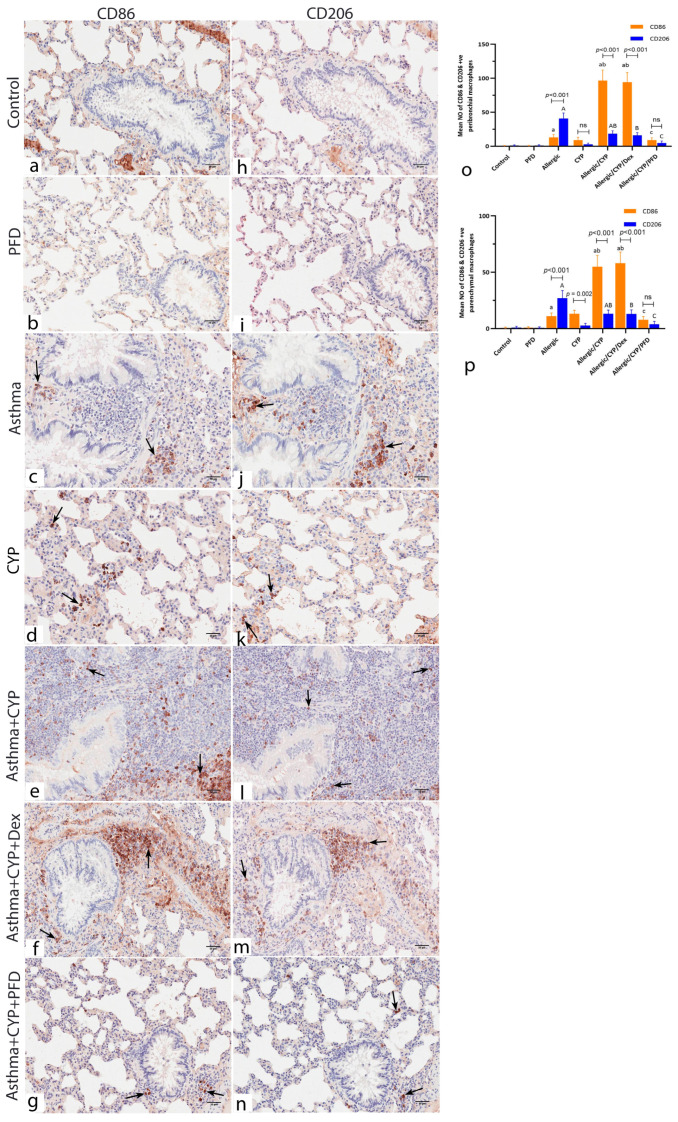
Immunoassay CD86 (**a**–**g**) and CD206 (**h**–**n**) stained serial lung sections showing the dominant macrophage phenotype in the corresponding areas of interest of each study group. (**a**,**h**) control and (**b**,**i**) PFD alone-treated groups show negative immunoreaction for both CDs markers. (**c**,**j**) Allergic pneumonitis group shows CD206-positive M2 macrophages (**j**) relatively numerous to the CD86-positive M1 cells. (**d**,**k)** CYP-exposed group shows multiple CD86-positive M1 macrophages (**d**) and scarce CD206-positive M2 macrophages (**k**). (**e**,**l**) Allergic/CYP-exposed group and (**f**,**m**) allergic/CYP/dexamethasone-treated group show numerous M1 and M2 macrophages with the predominance of the M1 phenotype (CD86 + ve cells). (**g**,**h**) Allergic/CYP/PFD-treated group shows very few positive M1 (**g**) and M2 (**h**) phenotype macrophages. (**o**,**p**) representative figures of the mean number of the peribronchial (**o**) and parenchymal (**p**) CD86 and CD206 +ve macrophages at the end of the experiment, *n* = 5. Regarding CD86 immunopositivity, (**a**) significant versus control (*p* < 0.001). (**b**) significant versus allergic pneumonitis group (*p* < 0.001). (**c**) significant versus allergic/CYP-exposed group (*p* < 0.001). The capital letter refers to the comparison between groups regarding the CD206. The horizontal lines indicate the statistical significance and the *p*-value between M1 and M2 distribution within the same group using post hoc Tukey’s two-way ANOVA. Scale bar: 50 µm (**a**–**n**).

**Table 1 cells-12-00994-t001:** PCR primers sequences and accession numbers of the studied genes.

Gene	Accession Number	Primer Sequence (5′ → 3′)
Signal transducer and activator of transcription 6 (*STAT6*)	NM_001044250.1	F: GAGCTACTGGTCAGATCGGC R: GGTTCCATCTGGCTCGTTGA
Mitogen-activated protein kinase p38 (*p38 MAPK*)	NM_031020.3	F: TCGGCACACTGATGACGAAA R: TCATGGCTTGGCATCCTGTT
Mucin 5AC (*MUC5AC*)	XM_039101269.1	F: GTTTCTGCACCATGTCAGGC R: TGGGGCGGTAGATGTGGATA
Interleukin 13 (*IL-13*)	NM_053828.1	F: TCTCGCTTGCCTTGGTGG R: CATTCAATATCCTCTGGGTCCTGT
Glyceraldehyde 3- phosphate dehydrogenase (*GAPDH*)	NM_001394060.2	F: GGTGCTGAGTATGTCGTGGAG R: ACAGTCTTCTGAGTGGCAGTGAT

F: forward, R: Reverse.

**Table 2 cells-12-00994-t002:** The effect of pirfenidone on the serum IgE and BALF levels of inflammatory cytokines, eosinophil, and neutrophil count in allergic pneumonitis/CYP-exposed rats at the end of the experiment.

	Serum IgE (ng/mL)	BALF TNF-α (pg/mL)	BALF IL13 (pg/mL)	BALF IL17 (pg/mL)	BALF INF-γ (pg/mL)	BALF Neutrophil Count/mm^3^	BALF Eosinophil Count/mm^3^
Control	12.7 ± 2.02	22.02 ± 2.2	37.1 ± 2.4	53.6 ± 4.9	8.7 ± 1.1	0.25 ± 0.5	0.38 ± 0.7
Pirfenidone (PFD)	13.1 ± 2.2	23.04 ± 2.9	37.3 ± 2.9	51 ± 3.7	9 ± 1.1	0.13 ± 0.4	0.63 ± 0.7
Allergic pneumonitis	95 ± 6.1 ^a (*p* < 0.01)^	83 ± 8.4 ^a (*p* < 0.01)^	184.5 ± 9 ^a (*p* < 0.01)^	90.2 ± 3.1 ^a (*p* < 0.01)^	11.8 ± 1.5 ^a (*p* = 0.01)^	0.75 ± 0.7	120.9 ± 6.2 ^a (*p* < 0.01)^
CYP	12.3 ± 2.3	58.9 ± 6.1 ^a (*p* < 0.01)^	44.7 ± 3.6	57.5 ± 5.6	12.4 ± 1.5 ^a (*p* =0.01)^	1 ± 0.8	1.9 ± 1.8
Allergic/CYP	83.7 ± 7.3 ^ab (*p* < 0.01)^	69.8 ± 9.4 ^ab (*p* < 0.01, = 0.003)^	132.4 ± 6.9 ^ab (*p* < 0.01)^	172.4 ± 16 ^ab (*p* < 0.01)^	18.9 ± 1.5 ^ab (*p* < 0.01)^	18.6 ± 4.3 ^ab (*p* < 0.01)^	15.5 ± 3.5 ^ab (*p* < 0.01)^
Allergic/CYP/Dex	77.9 ± 3.2	69.9 ± 8.3	123.9 ± 5.9	174 ± 12.8	19.7 ± 2	16.6 ± 5.2	11.5 ± 2.3
Allergic/CYP/PFD	18.6 ± 3.3 ^bc (*p* < 0.01)^	21.5 ± 3.7 ^bc (*p* < 0.01)^	36.2 ± 4.9 ^bc (*p* < 0.01)^	50.9 ± 4 ^bc (*p* < 0.01)^	9.9 ± 2.1 ^bc (*p* = 0.03, < 0.01)^	2.6 ± 1.6 ^c (*p* < 0.01)^	1 ± 0.8 ^bc (*p* < 0.01)^

The data are presented as a mean ± SD (*n* = 8 serum and 3 BALF samples). Superscript letters indicate the significant difference and the *p*-value between groups. ^a^: versus control, ^b^: versus allergic group, ^c^: versus allergic/CYP and allergic/CYP/Dex groups, using post hoc Tukey’s one-way ANOVA test.

**Table 3 cells-12-00994-t003:** The effect of pirfenidone on the lung tissue concentrations of MDA, SOD, and TNF-α in CYP-exposed allergic pneumonitis rats at the end of the experiment.

	MDA (µmol/mL)	SOD (mmol/min/mg)	TNF-α (pg/100 μg Protein)
Control	17.2 ± 2	96.9 ±9.4	11.9 ± 2.7
Pirfenidone (PFD)	17.4 ± 2.1	98.1 ± 7	12.9 ± 1.8
Allergic	34.7 ± 4.7 ^a (*p* < 0.01)^	25.6 ± 3.4 ^a (*p* < 0.01)^	19.3 ± 2.5 ^a (*p* = 0.01)^
CYP	42.1 ± 3.8 ^a (*p* < 0.01)^	34.8 ± 9.1 ^a (*p* < 0.01)^	32.2 ± 2.3 ^a (*p* < 0.01)^
Allergic/CYP	77.07 ± 4.2 ^ab (*p* < 0.01)^	12.8 ± 2.4 ^ab (*p* < 0.01, = 0.03)^	42.6 ± 3.6 ^ab (*p* < 0.01)^
Allergic/CYP/Dex	81.6 ± 8.3	14.8 ± 3.4	43.4 ± 1.1
Allergic/CYP/PFD	22.7 ± 5.4 ^bc (*p* < 0.01)^	73 ± 7.7 ^abc (*p* < 0.01)^	12.6 ± 2.8 ^bc (*p* = 0.004, < 0.01))^

The data are presented as a mean ± SD (*n* = 5 lung tissue samples). Superscript letters indicate the significant difference and the *p*-value between groups. ^a^ versus control, ^b^ versus allergic group, ^c^ versus allergic/CYP, and allergic/CYP/Dex groups, using post hoc Tukey’s one-way ANOVA test.

**Table 4 cells-12-00994-t004:** The effect of PFD on the peribronchial and perivascular mononuclear cellular infiltration in the lung tissues of the studied groups at the end of the experiment.

	Peribronchiolar Inflammation	Perivascular Inflammation
Control	0.7 ± 0.67	0.4 ± 0.52
Pirfenidone (PFD)	0.9 ± 0.74	0.8 ± 0.79
Allergic	1.4 ± 0.84 ^a (*p* = 0.03)^	1.2 ± 0.92 ^a (*p* = 0.04)^
CYP	0.9 ± 0.73	0.6 ± 0.69
Allergic/CYP	3.1 ± 0.99 ^ab (*p* < 0.001, = 0.001)^	2 ± 0.82 ^ab (*p* = 0.001, = 0.03)^
Allergic/CYP/Dex	2.8 ± 0.92 ^ab (*p* < 0.001, = 0.008)^	2.1 ± 0.88 ^ab (*p* = 0.001, = 0.01)^
Allergic/CYP/PFD	1.4 ± 0.97 ^c (*p* = 0..008)^	1.1 ± 0.99 ^c (*p* = 0.03)^

All data are presented as mean ± SD, *n* = 5. ^a^ significant versus control, ^b^ significant versus allergic group, ^c^ significant versus allergic/CYP & allergic/CYP/Dex groups. Statistical significance and the *p*-value are shown between the groups using one-way ANOVA and post hoc Tukey’s test.

## Data Availability

All data generated or analyzed in the current study were included in the published article and any further explanations are available upon reasonable request.

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
