# Peer review of "Modeling the Effects of Cypermethrin Toxicity on Ovalbumin-Induced Allergic Pneumonitis Rats: Macrophage Phenotype Differentiation and p38/STAT6 Signaling Are Candidate Targets of Pirfenidone Treatment"

_cells, 2023, doi:10.3390/cells12070994_

Round 1
Reviewer 1 Report
This is an interesting manuscript about the aggravating effect of oral cypermethrin (CYP) pesticide in allergic asthma, and the curative potential of pirfenidone (PDF), a synthetic pyridine. For that, the authors used the rat model of allergic pneumonitis. The authors concluded that “Pirfenidone improved the CYP-induced severe steroid-resistant asthma via the inhibition of the M1 macrophage differentiation through modulation of the STAT6/p38 MAPK pathway”. Of relevance is the effect of PDF on allergic asthma rats, which shows promising results. The aim and specific objects of the study are well stated, however some aspect regarding experimental design, results, English language, and discussion need to be improve.
Major concerns:
1- To confirm the statement that “Pirfenidone improvement of CYP-induced severe steroid resistance”, authors need to PDF treatment in allergic rat resistant to steroids, so 2 groups of experimentation are missed: the allergic pneumonitis with Dexamethasone (DEX) treatment, and the allergic/CYP/Dex+ post PDF treatment. Authors need to change the text, as the experimental design did not show PDF treatment in allergic rat resistant to steroids, but rather the effect of PDF in allergic asthma.
2- The authors indicate along the text that macrophages (Mg) in the lung change their phenotype based in the expression of p38 MAPK, CD86 and CD206. According to the procedure and results shown, most of the alveolar Mg must be washed away in the Bronchoalveolar lavage (BALF) process (section 2.6), before histology storage of the lungs. Information about the numbers of total cells in BALF and quantification and characterization of the alveolar Mg is relevant in these studies. Then, analyses of the alveolar Mg from BAFL, and comparison in the expression of these proteins need to be performed. The procedure for cell counting needs to explain the methodology. In fact, the histology pictures exceptionally show some alveolar macrophages. Also, to show M2 towards M1 change, authors need to show the expression CD86 associated to other markers.
3-The procedure for cell counting in the BALF needs to be explained in the methodology section. Also, pictures of cells from BALF and staining of the cells need to be shown.
4- Regarding p38 MAPK expression, the authors used this data to indicate higher greater infiltration, but to show that STAT6/p38 MAPK pathway, western blots from proteins extracts showing phosphorylation of these proteins will sustain the statement.
5- English language need to be revised, either expression and terms to improve the text of the results and discussion sections. Figure and table legends need to show more information, number of samples, times the experiments were performed, type of samples and technique used.
6- When showing the Statistical significance, the authors need to show the exact “p” number, indicating number of samples, and media+SD, between groups.
7- Procedure approval by ethical committee of the institution need to be provided.
8- Although few, there are studies in mice and rats about PFD and asthma, these need to be mentioned and discussed. For instance: Hirano et al., 2006; Mansoor et al., 2007.
Reviewer 2 Report
The authors describe a series of experiments to determine if pirfenidone (PFD) ameliorates the exacerbate pulmonary inflammation in a model of allergic pneumonitis in rats.
Major comments
The hypothesis is confusing and must be improved to facilitate the take-home-message. A good option might be to redirect it with a sentence authors use in the discussion: page 14, line 520, where authors state that PFD is an alternative to dexamethasone in steroid-resistant cases. In the beginning of the abstract and introduction, a part of explaining the importance of allergic pneumonitis, due to the use of the pesticide, they should also emphasize on steroid-resistance and that they are proposing pirfenidone as an alternative to this resistance. This is the hypothesis and the main aim of the study. To answer this, authors first develop a model of asthmatic immunoreactivity. Please introduce a clear hypothesis.
In the abstract introduce the problem in two lines, a proper hypothesis and the obtained conclusions from the results.
Dexamethasone should be mentioned in the introduction.
In the discussion start with the synopsis of the obtained results: page 13, line 458-line 462 and finish with a short conclusion (the take-home-message).
In the discussion introduce the pros and cons of the study.
From my point of view the results are more comprehensive if they are structured according to pathways/mechanisms of PFD (ex. Inflammation, macrophages…) than not structured on the impact of PFD on specific tissues/techniques.
The cellular count cannot be performed in the supernatant of the BALF (page 4, line 145).
Isn’t there lost information if first the BALF is perfomed and then one lung is used for lung tissue homogenates and the other is introduced in fixed? Alveolar macrophages and infiltrated cells will be in the BAL and you will be not able to detect them in the histologies. Are the histologies representative concerning macrophages?
For the hematoxylin eosin stainings, did you perform a score?
Minor comments
Introduce the consent of the Ethical Committee of the animal procedure.
How was OVA administered on days 0, 7 and 14? Page 3, line 115. Concerning the intranasal administration, it was daily from day 21 to 28th? Please specify.
CYP, PFD, and Dex were given daily form day 15 to 28? Intravenously or intranasal? If the animal received CYP and PFD it was administered together? And during the last week OVA was administered intranasal together with CYP, PFD and Dex? Please explain it in the methods section, maybe a graphical view of the timepoints of the model might help. How were the timings decided?
How were ELISA of cytokines corrected?
How was the extraction of RNA performed in lung tissue homogenates?
The resolution of figures 2, 3, 4 and 5 must be improved and bigger, the letter concerning each figure is too small and the same applies for the scale bar. The same applies for the corresponding graphs of the figures.
I would suppress the section 3.8 and introduce each measurement with the corresponding figure.
The morphometric study was performed by one expert? Isn’t it better two experts?
Exchange “rpm” for the international unit “x g”
Page 3 line 125: suppress “were allowed unlimited”.
In page 7 line 285, I would change “more decrease” for “lower levels”.
In page 8, line 311, replace “normal” for “healthy”.
In table the “MAD” should be redirected to “MDA”.
Reviewer 3 Report
Overall, this is a concise, clear, and good writing manuscript. The introduction is relevant and theory-based.The authors found that CYP exposure aggravated lung inflammation in OVA-induced rats asthma model and showed steroid-resistant changes, like p38 activation, neutrophil-mediated, M1 macrophage-related inflammation. They also found that the changes in allergic/CYP model were reversed by PFD, meanwhile not by dexamethasone treatment. On the mechanism, the auothers found the Pirfenidone improved the CYP-induced severe steroid-resistant asthma via the inhibition of the M1 macrophage differentiation through modulation of the STAT6/p38 MAPK pathway. Specific comments are as follows:
1. Figure2-5: The letters of the pannel are too small.
2. Figures: If change the histogram to dot plot will be better for see each value of the rats.
3. If a vitro experiment can be design and done for PFD inhibit the M1 macrophage differentiation through modulation of the STAT6/p38 MAPK pathway, that will be much better.
Round 2
Reviewer 1 Report
The authors answer the doubts raised, However, minor changes must be made for the publication of the article, as indicated below.
1-In discussion section, instead of indicating the hypotheses and conclusions in the third person, it would be more appropriate, and in accordance with the current scientific style, that the authors either use the first person, or indirect style, i.e.: lines 500-501, line 520; line 524; line 548; i.e.: the authors can use: we speculated, or our hypothesis, or the data suggested that…
2- When the authors commented on other one’s paper indicating only the reference number, it would be more appropriate to indicate the name of the author and then the references as in line 516. In fact, sometime is confusing (line 509), as one cannot distinguish whether the number is a reference of the previous line or to the next one. ie: lines 485, 488, 502,526
3- The authors need to Include the name of the Ethic committee and the number of the procedure approval in the animal section of the material and methods, as indicated in the instructions of the journal.
4-Table 2, there is a mistake in number typing Allergic pneumonitis/BALF -IFN “p=.01”,
5-. In section “2.8. Tissue homogenate analysis of TNF-α, MDA, SOD“ , the authors indicated the company and kit used, but they need to indicate in the text the method used, either PCR or ELISA, etc.
Author Response
The authors appreciate the extra remarks raised by the reviewers that will further improve the quality of the manuscript. All amendments were made according to the reviewer’s concerns and were introduced point by point. Amendments throughout the manuscript were highlighted with different colors. All those highlighted green indicated changes recommended by reviewer 1.
Reviewer #:1
The authors answer the doubts raised, However, minor changes must be made for the publication of the article, as indicated below.
1-In discussion section, instead of indicating the hypotheses and conclusions in the third person, it would be more appropriate, and in accordance with the current scientific style, that the authors either use the first person, or indirect style, i.e.: lines 500-501, line 520; line 524; line 548; i.e.: the authors can use: we speculated, or our hypothesis, or the data suggested that…
Response: edited
2- When the authors commented on other one’s paper indicating only the reference number, it would be more appropriate to indicate the name of the author and then the references as in line 516. In fact, sometime is confusing (line 509), as one cannot distinguish whether the number is a reference of the previous line or to the next one. ie: lines 485, 488, 502,526
Response: edited
3- The authors need to Include the name of the Ethic committee and the number of the procedure approval in the animal section of the material and methods, as indicated in the instructions of the journal.
Response: added
4-Table 2, there is a mistake in number typing Allergic pneumonitis/BALF -IFN “p=.01”,
Response: edited
5-. In section “2.8. Tissue homogenate analysis of TNF-α, MDA, SOD“ , the authors indicated the company and kit used, but they need to indicate in the text the method used, either PCR or ELISA, etc.
Response: edited
Reviewer 2 Report
The authors describe a series of experiments to determine if pirfenidone (PFD) ameliorates the exacerbate pulmonary inflammation in a model of allergic pneumonitis in rats. The authors have clearly improved the manuscript in the second round, however they have not addressed all the comments of the reviewers: according that in the manuscript the study is focused in macrophages, a proper explanation regarding the methodological concerns should had been introduced.
The main concern is that regarding the way samples were obtained, there is lost information, and although they inserted it at the end of the discussion, in the strengths and limitations section, it is not well addressed in the manuscript. Authors have now introduced that 3 animals were used for the BALF and 5 animals were used for the lung tissue homogenates and the immunohistochemical study. In the BALF they should address that after centrifugation the cells are counted in the pellet, not in the supernatant. More importantly, they should explain why in section 2.7, the lungs are first lavaged in saline, as if they do so there is lost information in the alveolar space, as they state in the limitations that the quantification and characterization of the alveolar macrophages were limited in the study. The explanation about why first the lungs are washed with cooled saline should be introduced in the methodology and supported by literature, and the consequences of using this methodology should also be explained in the discussion, not only in the limitations section, as it is an important parameter, especially regarding that they are studying macrophages.
In addition, the strengths and limitations introduced are quite vague. I would recommend to insert them before the last conclusion. As a strength I would introduce the development of the new model; the OVA-sensitization/CYP exposure model, that completely reproduces the injury parameters. As a limitation, I do not understand why authors say that it was not conclusive the combination of PFD and dexamethasone, as it is not the aim of the study, they do not perform studies for that. However, they can include that it would give further information. Regarding the macrophages, authors should state why alveolar macrophages quantification and characterization are limited in the current study.
In the abstract, authors should address that allergic pneumonitis was induced by two week intraperitoneal OVA injections, according to the graph and methods section.
In the methods, page 3 line 129, suppress were allowed free access. Ad libitium already means free or unlimited.
The number of the Ethical Committee should be inserted in the methods section (only the number).
Author Response
The authors appreciate the extra remarks raised by the reviewers that will further improve the quality of the manuscript. All amendments were made according to the reviewer’s concerns and were introduced point by point. Amendments throughout the manuscript were highlighted with different colors. All those highlighted yellow indicated changes recommended by reviewer 2.
Reviewer #:2
- The authors describe a series of experiments to determine if pirfenidone (PFD) ameliorates the exacerbate pulmonary inflammation in a model of allergic pneumonitis in rats. The authors have clearly improved the manuscript in the second round, however they have not addressed all the comments of the reviewers: according that in the manuscript the study is focused in macrophages, a proper explanation regarding the methodological concerns should had been introduced. The main concern is that regarding the way samples were obtained, there is lost information, and although they inserted it at the end of the discussion, in the strengths and limitations section, it is not well addressed in the manuscript. Authors have now introduced that 3 animals were used for the BALF and 5 animals were used for the lung tissue homogenates and the immunohistochemical study. In the BALF they should address that after centrifugation the cells are counted in the pellet, not in the supernatant.
Response: edited
- More importantly, they should explain why in section 2.7, the lungs are first lavaged in saline, as if they do so there is lost information in the alveolar space, as they state in the limitations that the quantification and characterization of the alveolar macrophages were limited in the study. The explanation about why first the lungs are washed with cooled saline should be introduced in the methodology and supported by literature, and the consequences of using this methodology should also be explained in the discussion, not only in the limitations section, as it is an important parameter, especially regarding that they are studying macrophages.
Response: edited and written in clear way. In section2.7, the lungs were not lavaged, just slightly washed (rinsed) by cooled (to keep the tissue moist and vital) saline as recommended by the Lab to remove blood clots and other tissue remnants, then immersed in the fixative for 48 hrs. Lung lavage was not performed in animals whose lung samples were used for histology and immunohistochemistry.
- In addition, the strengths and limitations introduced are quite vague. I would recommend to insert them before the last conclusion. As a strength I would introduce the development of the new model; the OVA-sensitization/CYP exposure model, that completely reproduces the injury parameters. As a limitation, I do not understand why authors say that it was not conclusive the combination of PFD and dexamethasone, as it is not the aim of the study, they do not perform studies for that. However, they can include that it would give further information.
Response: edited in the text
- Regarding the macrophages, authors should state why alveolar macrophages quantification and characterization are limited in the current study.
Response: The characterization & quantification of macrophages were conducted in the immunohistochemistry-stained slides, while not in the BALF samples as it was not designed as a part of the used methods (specified in the last part of the discussion). The main goal was to investigate the cell-type mediated asthma (whether eosinophil or neutrophil), and this was confirmed in BALF and histologies. However, it will be consistent and compliant if the BALF samples were analyzed for macrophages.
- In the abstract, authors should address that allergic pneumonitis was induced by two week intraperitoneal OVA injections, according to the graph and methods section.
Response: edited
In the methods, page 3 line 129, suppress were allowed free access. Ad libitium already means free or unlimited.
Response: edited
- The number of the Ethical Committee should be inserted in the methods section (only the number).
Response: edited. Reviewer 1 recommended to write both the name and number